# Formal Representation and Solution of Plane Geometric Problems

**Xiaokai Zhang, Na Zhu, Cheng Qin, Yang Li, Tuo Leng**[*]
School of Computer Engineering and Science
Shanghai University
tleng@shu.edu.cn

**Zhenbing Zeng**
College of Sciences
Shanghai University

**Project:** `https://github.com/FormalGeo/FormalGeo`

## Abstract

Geometric problem solving has always been a long-standing challenge in the fields of mathematical reasoning and artificial intelligence. In this paper, we present formalgeo7k, a geometric problem dataset based on rigorous geometry formalization theory and consistent geometry formal system, serving as a benchmark for various tasks such as geometric diagram parsing and geometric problem solving. All problems are annotated with problem text, problem diagram, formal descriptions, and solution. Combining symbolic solver and deep learning techniques, we can achieve human-like, traceable and explainable solutions, which are stored in a hypergraph for graph-related tasks. We experiment with various methods and the best method achieves only 86.40% on formalgeo7k. This shows that formalgeo7k presents a challenge for future research.

## 1 Introduction

Geometric problem solving (GPS) has always been a long-standing challenge [22, 30, 13] in the fields of mathematical reasoning and artificial intelligence, owing to the cross-modal forms of knowledge and the absence of automated solving methods. A typical geometric problem consists of a textual problem description and a geometric diagram. GPS requires solvers to possess multimodal fusion and reasoning capabilities, which have attracted much attention recently. Existing works mostly construct a neuro-symbolic system for GPS. Deductive Database (DD) methods [14, 40, 24] parse the problem text and diagram into formal language and then solve the problem by logical reasoning. Program sequence generation (PSG) methods [5, 36, 29] encode the problem text and diagram, input the encoding into a decoder, and generate a program sequence, which is then executed by a program executor.

However, existing methods focus on the research of the neural part while neglecting the symbolic part. First, existing methods fail to achieve a human-like problem-solving process. DD methods cannot eliminate redundant theorems, and PSG methods lack mathematical rigor. This not only undermines the readability of the solutions but also limits their application in mathematics education. Second, existing methods lack research on geometry formalization theory. This not only fails to ensure the consistency of the solver's reasoning process but also hinders the expansion of symbolic systems. Defining new theorems requires modifying the solver's code, making it difficult to represent more complex problems, such as problems at International Mathematical Olympiad (IMO) level. Furthermore, existing datasets are small in scale and cannot serve as benchmarks for training and evaluating large language model (LLMs). Some datasets are poorly annotated or contain errors, making them unsuitable as a unified benchmark.

---

[*]Corresponding Author

38th Conference on Neural Information Processing Systems (NeurIPS 2024) Workshop on MATH-AI.

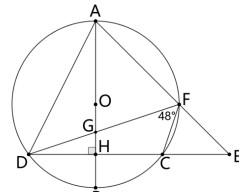

**Image CDL**
Equal(MeasureOfAngle(CFD),48)
PerpendicularBetweenLine(DH,GH)

**Text CDL**
Equal(MeasureOfAngle(CFD),48)
IsCentreOfCircle(O,O)
IsDiameterOfCircle(AB,O)
PerpendicularBetweenLine(DH,GH)

**Construction CDL**
Shape(OAD,DA)
Shape(AD,DG,GO,OA)
Shape(AO,OG,GF,FA)
Shape(OFA,AF)
Shape(GD,DH,HG)
Shape(FG,GH,HC,CF)
Shape(OCF,FC)
Shape(OCF,CE,EF)

Shape(BH,HD,ODB)
Shape(CH,HB,OBC)
Collinear(AOGHB)
Collinear(DGF)
Collinear(AFE)
Collinear(DHC)
Cocircular(O,ADBCF)

**Problem Text**
English: As shown in the diagram, ∠CFD=48°, O is the center of
⊙O, the diameter of ⊙O is AB, DH is perpendicular to GH. Find the
measure of ∠EFC.
Chinese: 如图所示，∠CFD=48°，O是⊙O的圆心，⊙O的直径为
AB，DH垂直于GH。求∠EFC的大小。

**Image Annotation (parsed from .gbb)**
Points: {A:[919,980], B:[1378,827], C:[919,1103], ..., O:[919,827]}

**Goal CDL**              **Answer**
Value(MeasureOfAngle(EFC))   66

**Theorem Sequences**
circle_property_chord_perpendicular_bisect_arc(1,ODC,OHB)
arc_addition_length(1,ODB,OBC)
similar_arc_judgment_cocircular(1,ODB,ODC)
similar_arc_property_length_ratio(1,ODB,ODC)
similar_arc_property_measure_ratio(1,ODB,ODC)
arc_property_circumference_angle_external(1,ODB,A)
arc_property_circumference_angle_external(1,OAC,F)
arc_property_circumference_angle_external(1,ODC,F)
triangle_property_angle_sum(1,ADH)
arc_property_circumference_angle_internal(1,OAC,D)
adjacent_complementary_angle(1,EFC,CFA)

Figure 1: An example of annotated geometric problem in formalgeo7k.

To address these issues, we develop FormalGeo, a consistent geometry formal system based on rigorous geometry formalization theory [37]. We further construct the formalgeo7k dataset, which contains 7,000 geometric problems. All problems are annotated with problem text, problem diagrams, formal descriptions, and solutions, as shown in Fig. 1. This dataset can serve as a benchmark for various tasks, such as GPS, geometric diagram parsing (GDP), geometric relation extraction (GRE), geometric problem formalization and the evaluation of LLMs reasoning abilities.

To assess the dataset difficulty, we experiment with several methods and the best method only achieves 86.40% problem solving success rate (PSSR). This shows that formalgeo7k presents a challenge for future research.

## 2 formalgeo7k dataset

We present formalgeo7k, a geometric problem dataset based on rigorous geometry formalization theory and consistent geometry formal system, serving as a benchmark for various tasks such as geometric diagram parsing and geometric problem solving. All code and datasets are open-source and available at `https://github.com/FormalGeo/FormalGeo`. Refer to Appx. D for instructions on how to use *pip* to build FormalGeo formal systems and download formalgeo7k.

### 2.1 Geometry formal system

We build FormalGeo, a consistent geometry formal system based on rigorous geometry formalization theory [37]. FormalGeo incorporates diagrammatic reasoning, algebraic computation, relational reasoning, and logical operations into a unified framework of geometry predicate logic, whose syntax is similar to first-order predicate logic. Within this formal framework, we can transform geometric diagrams and text into a consistent formal language, which not only maintains good readability but can also be mechanically processed by computer. The final solution of geometric problems can be represented as hypergraph, where conditions serve as hypernodes and theorems as hyperedges, thus transforming into a human-like solution. In addition, FormalGeo uses geometry definition language (GDL) to define new predicates or theorems and uses condition declaration language (CDL) to describe geometry problems, making it extremely convenient to extend the formal system and represent complex problems. Details of GDL are provided in Appx. A. A formal representation and solution of a 2022 IMO geometry problem can be found in Appx. B.

## 2.2 Dataset annotation

Our formalgeo7k Dataset contains 7,000 plane geometric problems, sourced from the Geometry3K [14] (40.69%), GeoQA [5] (53.38%), and GeoQA+ [3] (2.93%). Our annotation task can be divided into four parts: 1. Standardizing the style of problem texts (English and Chinese). For problems lacking English or Chinese descriptions, we add the missing descriptions. 2. Redrawing geometric diagrams using GeoGebra and saving the *.gbb* files, which contains detailed positional information of the geometric elements. 3. Annotating geometric diagrams and texts using FormalGeo formal language. 4. Adding the theorem sequences required to solve the geometric problems.

Once the problems are annotated, we assign a different annotator to review and verify the correctness of the annotations. Finally, all problems are input into a symbolic solver FGPS [38] to check the correctness of the syntax, ensure that the annotated theorems can solve the problem and eliminate redundant theorems. By treating the intermediate results of a problem as a new goal, we can decompose a geometric problem into multiple sub-problem, automatically expanding the number of problems to tens of thousands. This results in a dataset with a smoother difficulty curve, and the number of problems becomes sufficient for training and evaluating LLMs.

Sixteen trained graduate master's students participated in the dataset annotation tasks. The annotation and reviewing process took approximately 1,500 person-hours.

## 2.3 Dataset Statistics

Our dataset has 7,000 geometric problems. Each problem text (English) has a maximum of 444 characters and a minimum of 58 characters, with an average problem text length of 114.22. The construction CDL describes the topological structure of a geometric problem, and the number of its statements roughly reflects the complexity of the geometric diagram. The average length of construction CDL statements is 5.97. Both text CDL and image CDL describe the conditions of the geometric problem, with the average number of statements being 4.02 and 2.87, respectively. In the text CDL, the most frequently occurring predicates are *Equal* (61.04%), *IsCentreOfCircle* (9.67%), *PerpendicularBetweenLine* (8.69%), *ParallelBetweenLine* (4.92%), and *IsTangentOfCircle* (3.22%). In the diagram CDL, The proportions of *Equal*, *PerpendicularBetweenLine*, and *ParallelBetweenLine* are 80.95%, 12.16%, and 6.89%, respectively. The number of theorems required for GPS serves as an measure of the difficulty of the problem. Each problem involves a maximum of 28 theorems and a minimum of 1 theorems, with an average theorem number of 4.34. The detailed statistical information can be found in Appx. C.

# 3 Experiments

## 3.1 Benchmark methods

We tested several methods on formalgeo7k. It is important to note that most SOTA models for GPS are trained in specific symbolic environments (such as those provided by Geometry3K [14] or GeoQA [5]), and adapting them to the FormalGeo would require significant manual effort. Therefore, we only compare methods using the FormalGeo environment. All methods use annotated text CDL and image CDL as input to the model. Parsing CDL from the original problem diagram and text is still a challenge, and we leave it for future work.

**Forward Search** [38]. This is a purely symbolic approach. It starts from the initial conditions of the problem and continuously apply theorems to derive new conditions until the goal is achieved. We run the forward search method using four different strategies (breadth-first, depth-first, random and beam) and presented the results of the best strategy (random). The maximum search depth is set to 15, the beam size to 20, and the timeout for each problem is set to 600 seconds.

**Backward Search** [38]. This is a purely symbolic approach. It begins with the problem-solving goal, expands it into multiple sub-goals, and repeats this process until all sub-goals are resolved. The best strategy for the backward search is breadth-first, with the other parameters set the same as in the forward search. We run the search method on two Intel i9-10900X processors, one AMD Ryzen 9 5900X, and one AMD Ryzen 9 7950X, using multiple processes while maintaining a CPU utilization rate of 80%. The total duration of the search is approximately 3 days.

Table 1: Details of PSSR.

| Method | Total | $L_1$ | $L_2$ | $L_3$ | $L_4$ | $L_5$ | $L_6$ |
|---|---|---|---|---|---|---|---|
| Forward Search | 39.71 | 59.24 | 40.04 | 33.68 | 16.38 | 5.43 | 4.79 |
| Backward Search | 35.44 | 67.22 | 33.72 | 11.15 | 6.67 | 6.07 | 1.03 |
| FGeo-TP | 80.86 | 96.43 | 85.44 | 76.12 | 62.26 | 48.88 | 29.55 |
| FGeo-DRL | 86.40 | 97.65 | 94.21 | 85.87 | 70.45 | 46.81 | 32.18 |
| HyperGNet | 85.53 | 95.44 | 89.46 | 84.25 | 77.84 | 50.00 | 45.76 |

**FGeo-TP** [9]. This method utilizes the language model to predict the theorem sequences for GPS, reducing the search complexity. We chose BART-base [10] as the theorem predictor. The training epochs were set to 20, with an initial learning rate of $3 \times 10^{-5}$. After theorem prediction, we ran the backward search method using a random strategy.

**FGeo-DRL** [40]. This method leverages a pre-trained natural language model to establish a policy network for theorem selection and employ monte carlo tree search for heuristic exploration. We chose DistilBERT [21] as the policy network to learn how to select a theorem from a 234 action space for the current problem step. The implementation details and training methods are consistent with those in the original paper.

**HyperGNet** [39]. This method builds a neural-symbolic system to effectively embed geometry knowledge and automatically perform human-like geometric problem solving. We train HyperGNet on a single GeForce RTX 4090. During the model's training phase, we optimize the model parameters using the Adam algorithm, with a learning rate of $10^{-5}$, batch size of 64 and training epochs of 50. Executing a single training session of the neural network only require approximately 30 minutes.

### 3.2 Experimental results

To provide a more detailed comparison of different models' capabilities, we divided the dataset into 6 levels based on the length of the theorem sequence $l$ required to solve the problem: $L_1(l \leq 2)$, $L_2(3 \leq l \leq 4)$, $L_3(5 \leq l \leq 6)$, $L_4(7 \leq l \leq 8)$, $L_5(9 \leq l \leq 10)$, $L_6(l \geq 11)$. The experimental results are shown in Tab. 1.

It is evident that the longer the theorem sequence required to solve the problem, the higher the difficulty and the lower PSSR. We can see that, compared to traditional search methods, heuristic search combined with deep learning techniques has significantly improved the problem-solving success rate. FGeo-DRL achieved the highest overall PSSR, but its performance on solving difficult problems was lacking. HyperGNet, while slightly behind FGeo-DRL in overall PSSR, performed better on more challenging problems.

## 4 Related Work

The study of GPS has a long history, which can be broadly divided into algebraic methods and synthetic methods. Algebraic methods transform geometric problems into a system of algebraic equations consisting of polynomials and inequalities, such as Wu's method [27], Gröbner basis methods [2] and elimination methods [26]. Synthetic methods encompass a wider range of approaches, including search-based methods [8, 16], knowledge-based methods [7], geometric invariants-based methods [34, 6], and machine Learning and optimization methods [23, 1, 20]. GPS has seen further advancements in recent years. Existing methods predominantly integrate deep learning and symbolic reasoning to construct a neuro-symbolic system for solving geometric problems. DD methods parse the problem text and diagram into formal language and then solve the problem by logical reasoning. Representative DD methods include Inter-GPS [14], GeoDRL [18], AlphaGeometry [24], FGeo [40, 9] and E-GPS [28]. PSG methods encode the problem text and diagrams, input the encoding into a decoder, and generate a program sequence, which is then executed by a program executor. Representative PSG methods include NGS [5], Geoformer [4], DPE [3], PGPSNet [36], SAC-GPS [17], UniMath [12], LANS [11], DualGeoSolver [29] and adaptive learning model [32]. Several geometry formal systems and datasets have been developed. We compared existing datasets with formalgeo7k, as shown in Tab. 2. In addition to geometric problem solving, tasks such as geometric diagram parsing [35] and geometric formalization [15] have also begun to attract increasing attention.

Table 2: Comparative analysis with existing geometric problem datasets.

| Method | Task | Size | Comparative Metrics for GPS task | | | | | | | |
|---|---|---|---|---|---|---|---|---|---|---|
| | | | FM | FW | BW | NT | PT | FS | HS | CR |
| GEOS [23] | GPS | 186 | DD | ✓ | | ✓ | | | | |
| GEOS++ [19] | GPS | 1,406 | DD | ✓ | | ✓ | | | | |
| GEOS-OS [20] | GPS | 2,235 | DD | ✓ | | ✓ | | | | |
| Geometry3K [14] | GPS | 3,002 | DD | ✓ | | ✓ | | | | |
| GeoQA [5] | GPS | 5,010 | PSG | ✓ | | ✓ | | | | |
| GeometryQA [25] | GPS | 1,398 | PSG | ✓ | | ✓ | | | | |
| GeoRE [31] | GRE | 1,398 | - | - | - | - | - | - | - | - |
| GeoQA+ [3] | GPS | 7,528 | PSG | ✓ | | ✓ | | | | |
| UniGeo [4] | GPS | 14,541 | PSG | ✓ | | ✓ | ✓ | | | |
| PGDP5K [35] | GDP | 5,000 | - | - | - | - | - | - | - | - |
| PGPS9K [36] | GPS | 9,022 | PSG | ✓ | | ✓ | | | | |
| GeoEval [33] | GPS | 5050 | - | ✓ | | ✓ | | | | ✓ |
| formalgeo7k | All | 7000 | DD | ✓ | ✓ | ✓ | ✓ | ✓ | ✓ | ✓ |

\* All denotes GPS+GDP+GRE. The 8 comparative metrics are: Formalization Methods, ForWard solving, BackWard solving, Numerical Targets, Proving Targets, Formal System, Human-like Solutions and Complexity Ratings.

## 5 Conclusion

Based on rigorous geometry formalization theory and a consistent geometry formal system, we developed formalgeo7k, a dataset containing 7,000 annotated geometry problems, including problem text, problem diagrams, formal descriptions, and solutions. formalgeo7k serves as a benchmark for various tasks such as geometric diagram parsing and geometric problem solving, directly benefiting both AI4MATH and AI4EDU research. Experimental results indicate that formalgeo7k presents a challenge for future research.

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

Table 3: Detailed statement examples for defining a predicate.

| name | item | content |
|---|---|---|
| IsMidpointOfLine(M,AB) | ee_check | Point(M) |
| | | Line(AB) |
| | | Collinear(AMB) |
| | fv_check | M,AB |
| | multi | M,BA |
| | extend | Equal(LengthOfLine(AM),LengthOfLine(MB)) |
| LengthOfLine(AB) | ee_check | Line(AB) |
| | sym | ll |
| | multi | BA |

# A    Details of GDL

FormalGeo formal languages are categorized into geometry definition language (GDL) and condition declaration language (CDL). The former is used to define relations, attributes, theorems, and other elements of a geometry formal system, while the latter is employed for declaring conditions and goals in geometric problems. This chapter introduces GDL. An example of CDL can be found in Appx. B.

GDL can be divided into predicate definition language and theorem definition language.

## A.1    Predicate definition language

Predicate definition language is used to define geometric relations and geometric attributions. FormalGeo comprises 89 predicates, including 25 fundamental predicates (Tab. 4) built into the solver and 12 entities (Tab. 5), 31 entity relationships (Tab. 6), and 21 attributions (Tab. 7) defined using the predicate definition language.

The detailed statements for defining a predicate is as shown in the Tab. 3, including the predicate name and point variable declaration, validity check declaration, multiple representations, and automatic expansion. Additionally, when defining attributes, it also includes symbolic form declaration.

## A.2    Theorem definition language

Theorems are defined using the geometry predicate logic, comprising two parts: premises and conclusions, as shown in the Tab. 8. FormalGeo defines 200 theorems.

# B    An example of annotated IMO geometry problem

Based on rigorous geometry formalization theory and a carefully developed symbolic solver, FormalGeo can represent, verify, and solve IMO-level geometry problems. Fig. 2 shows the original problem from the 2022 IMO Problem 4 and its formalized description.

The final solution of geometric problems can be represented as a hypergraph, where conditions serve as hypernodes and theorems as hyperedges, as shown in Fig. 3. For this hypergraph, we can easily apply rule-based methods to transform it into a human-like solution, as illustrated in Fig. 4. We can also use LLMs to convert it into a more readable solution, which we leave for future research.

Table 4: Predicates built into the solver.

| id | type | name | examples |
|----|------|------|----------|
| 1 | Construction | Shape | Shape(AB,BC,CA) |
| 2 | Construction | Collinear | Collinear(ABCD) |
| 3 | Construction | Cocircular | Cocircular(O,ABC) |
| 4 | BasicEntity | Point | Point(A) |
| 5 | BasicEntity | Line | Line(AB) |
| 6 | BasicEntity | Arc | Arc(OAB) |
| 7 | BasicEntity | Angle | Angle(ABC) |
| 8 | BasicEntity | Polygon | Polygon(ABCD) |
| 9 | BasicEntity | Circle | Circle(O) |
| 10 | Algebra | Equal | Equal(a,b) |
| 11 | Algebra | Equation | Equation(a-b) |
| 12 | Attribution | Free | Free(x) |
| 13 | Operation | Add | Equal(Add(a,b,c),1) |
| 14 | Operation | Sub | Equal(Sub(a,b),1) |
| 15 | Operation | Mul | Equal(Mul(a,b,c),1) |
| 16 | Operation | Div | Equal(Div(a,b),1) |
| 17 | Operation | Pow | Equal(Pow(a,b),1) |
| 18 | Operation | Mod | Equal(Mod(a,b),1) |
| 19 | Operation | Sqrt | Equal(Sqrt(a),1) |
| 20 | Operation | Sin | Equal(Sin(a),1/2) |
| 21 | Operation | Cos | Equal(Cos(a),1/2) |
| 22 | Operation | Tan | Equal(Tan(a),1) |
| 23 | Target | Value | Value(a) |
| 24 | Target | Equal | Equal(a,b) |
| 25 | Target | Relation | Relation(RightTriangle(ABC)) |

Table 5: Entities defined using the predicate definition language.

| id | type | examples |
|----|------|----------|
| 26 | Entity | RightTriangle(ABC) |
| 27 | Entity | IsoscelesTriangle(ABC) |
| 28 | Entity | IsoscelesRightTriangle(ABC) |
| 29 | Entity | EquilateralTriangle(ABC) |
| 30 | Entity | Kite(ABCD) |
| 31 | Entity | Parallelogram(ABCD) |
| 32 | Entity | Rhombus(ABCD) |
| 33 | Entity | Rectangle(ABCD) |
| 34 | Entity | Square(ABCD) |
| 35 | Entity | Trapezoid(ABCD) |
| 36 | Entity | IsoscelesTrapezoid(ABCD) |
| 37 | Entity | RightTrapezoid(ABCD) |

Table 6: Relations defined using the predicate definition language.

| id | type | examples |
| --- | --- | --- |
| 38 | Relation | IsMidpointOfLine(M,AB) |
| 39 | Relation | IsMidpointOfArc(M,OAB) |
| 40 | Relation | ParallelBetweenLine(AB,CD) |
| 41 | Relation | PerpendicularBetweenLine(AC,BC) |
| 42 | Relation | IsPerpendicularBisectorOfLine(AB,CD) |
| 43 | Relation | IsBisectorOfAngle(BD,ABC) |
| 44 | Relation | IsMedianOfTriangle(AD,ABC) |
| 45 | Relation | IsAltitudeOfTriangle(AD,ABC) |
| 46 | Relation | IsMidsegmentOfTriangle(DE,ABC) |
| 47 | Relation | IsCircumcenterOfTriangle(O,ABC) |
| 48 | Relation | IsIncenterOfTriangle(O,ABC) |
| 49 | Relation | IsCentroidOfTriangle(O,ABC) |
| 50 | Relation | IsOrthocenterOfTriangle(O,ABC) |
| 51 | Relation | CongruentBetweenTriangle(ABC,DEF) |
| 52 | Relation | MirrorCongruentBetweenTriangle(ABC,DEF) |
| 53 | Relation | SimilarBetweenTriangle(ABC,DEF) |
| 54 | Relation | MirrorSimilarBetweenTriangle(ABC,DEF) |
| 55 | Relation | IsAltitudeOfQuadrilateral(EF,ABCD) |
| 56 | Relation | IsMidsegmentOfQuadrilateral(EF,ABCD) |
| 57 | Relation | IsCircumcenterOfQuadrilateral(O,ABCD) |
| 58 | Relation | IsIncenterOfQuadrilateral(O,ABCD) |
| 59 | Relation | CongruentBetweenQuadrilateral(ABCD,EFGH) |
| 60 | Relation | MirrorCongruentBetweenQuadrilateral(ABCD,EFGH) |
| 61 | Relation | SimilarBetweenQuadrilateral(ABCD,EFGH) |
| 62 | Relation | MirrorSimilarBetweenQuadrilateral(ABCD,EFGH) |
| 63 | Relation | CongruentBetweenArc(OAB,OCD) |
| 64 | Relation | SimilarBetweenArc(OAB,OCD) |
| 65 | Relation | IsDiameterOfCircle(AB,O) |
| 66 | Relation | IsTangentOfCircle(PA,O) |
| 67 | Relation | IsCentreOfCircle(P,O) |
| 68 | Relation | ConcyclicBetweenPoints(A,B,C,D) |

Table 7: Attributions defined using the predicate definition language.

| id | type | examples |
| --- | --- | --- |
| 69 | Attribution | LengthOfLine(AB) |
| 70 | Attribution | MeasureOfAngle(ABC) |
| 71 | Attribution | PerimeterOfTriangle(ABC) |
| 72 | Attribution | AreaOfTriangle(ABC) |
| 73 | Attribution | HeightOfTriangle(ABC) |
| 74 | Attribution | RatioOfSimilarTriangle(ABC) |
| 75 | Attribution | RatioOfMirrorSimilarTriangle(ABC) |
| 76 | Attribution | PerimeterOfQuadrilateral(ABCD) |
| 77 | Attribution | AreaOfQuadrilateral(ABCD) |
| 78 | Attribution | HeightOfQuadrilateral(ABCD) |
| 79 | Attribution | RatioOfSimilarQuadrilateral(ABCD) |
| 80 | Attribution | RatioOfMirrorSimilarQuadrilateral(ABCD) |
| 81 | Attribution | LengthOfArc(OAB) |
| 82 | Attribution | MeasureOfArc(OAB) |
| 83 | Attribution | RatioOfSimilarArc(OAB) |
| 84 | Attribution | RadiusOfCircle(O) |
| 85 | Attribution | DiameterOfCircle(O) |
| 86 | Attribution | PerimeterOfCircle(O) |
| 87 | Attribution | AreaOfCircle(O) |
| 88 | Attribution | PerimeterOfSector(OAB) |
| 89 | Attribution | AreaOfSector(OAB) |

Table 8: Detailed statement examples for defining a theorem.

| name | item | content |
|---|---|---|
| midpoint_of_line_ judgment(M,AB) | premise | Collinear(AMB)& Equal(LengthOfLine(AM),LengthOfLine(MB)) |
| | conclusion | IsMidpointOfLine(M,AB) |
| vertical_angle (AOC,BOD) | premise | Collinear(AOB)&Collinear(COD)& Angle(AOC)&Angle(BOD) |
| | conclusion | Equal(MeasureOfAngle(AOC),MeasureOfAngle(BOD)) |

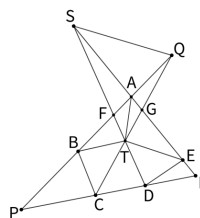

**Problem Text**

English: As shown in the figure, BC=DE, TB=TD, TC=TE, and ∠ABT=∠TEA. Prove that points P, R, Q, and S are concyclic.

Chinese: 如图所示，BC=DE，TB=TD，TC=TE，∠ABT=∠TEA。求证P、R、Q、S四点共圆。

**Text CDL**
Equal(LengthOfLine(BC),LengthOfLine(DE))
Equal(LengthOfLine(TB),LengthOfLine(TD))
Equal(LengthOfLine(TC),LengthOfLine(TE))
Equal(MeasureOfAngle(ABT),MeasureOfAngle(TEA))

**Goal CDL**
Relation(ConcyclicBetweenPoints(P,R,Q,S))

**Construction CDL**
Shape(SA,AQ,QS)       Shape(GT,TE,EG)       Collinear(SAGER)
Shape(SF,FA,AS)       Shape(BP,PC,CB)       Collinear(SFTD)
Shape(QA,AG,GQ)       Shape(BC,CT,TB)       Collinear(PBFAQ)
Shape(AF,FT,TA)       Shape(TC,CD,DT)       Collinear(PCDR)
Shape(AT,TG,GA)       Shape(TD,DE,ET)       Collinear(CTGQ)
Shape(FB,BT,TF)       Shape(ED,DR,RE)

**Theorem Sequences**
congruent_triangle_judgment_sss(1,TBC,TDE)
congruent_triangle_property_angle_equal(1,TBC,TDE)
adjacent_complementary_angle(1,CTB,BTQ)
adjacent_complementary_angle(1,STE,ETD)
mirror_similar_triangle_judgment_aa(1,QBT,STE)
mirror_similar_triangle_property_line_ratio(1,QBT,STE)
mirror_similar_triangle_property_line_ratio(1,BTQ,EST)
concyclic_between_points_judgment_circular_power(1,STD,CTQ)
concyclic_between_points_property_angle_equal(1,C,D,Q,S)
mirror_similar_triangle_property_angle_equal(1,QBT,STE)
triangle_property_angle_sum(1,QPC)
adjacent_complementary_angle(1,PCQ,QCR)
angle_addition(1,QSA,ASF)
concyclic_between_points_judgment_sum_of_angles(2,P,R,Q,S)

Figure 2: An example of annotated geometry problem (2022 IMO Problem 4).

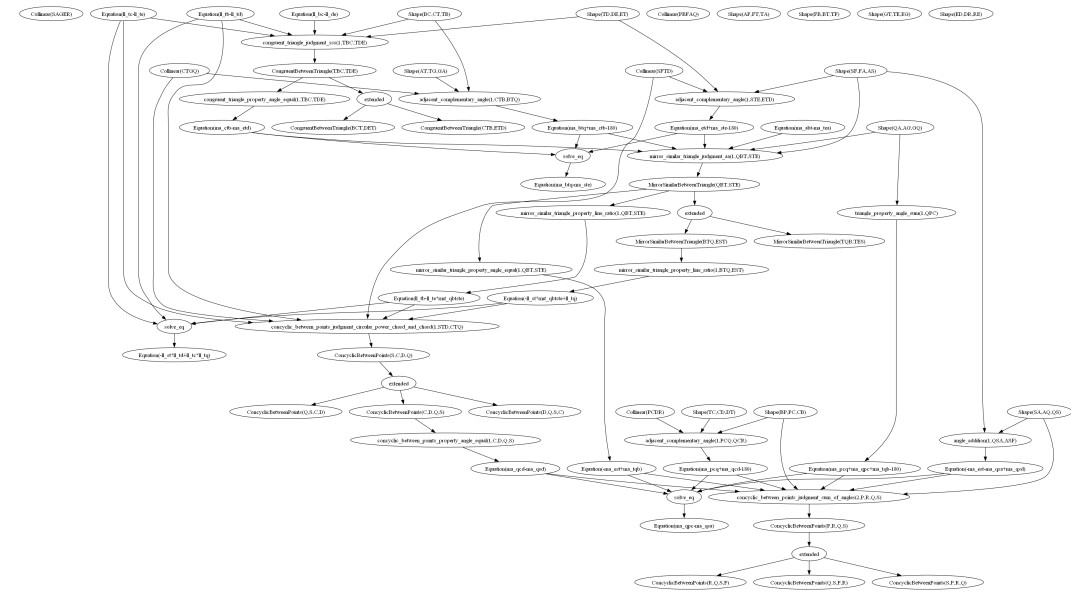

Figure 3: Solution hypergraph generated by symbolic solver FGPS [38] (2022 IMO Problem 4).

**Human-like Solution**

From the conditions of the problem, we have $\angle ABT = \angle TEA$ (1), BC=DE (2), TB=TD (3), TC=TE (4);

By the theorem of adjacent supplementary angles, $\angle ETD = -\angle STE + 180$ (5);

Given conditions (2)(3)(4), by the triangle congruence theorem (SSS), we have triangles TBC and TDE are congruent (6);

Given conditions (6), by the property of congruent triangles (equal angles), we have $\angle CTB = \angle ETD$ (7);

By the theorem of adjacent supplementary angles, we have $\angle BTQ = -\angle CTB + 180$ (8);

Given conditions (5)(1)(7)(8), by the similarity theorems for triangles (AA), we have triangles QBT and STE are mirror-image similar triangles(9);

Given conditions (9), by the the property of similar triangles, TB=TE×Ratio($\triangle$QBT,$\triangle$STE) (10);

Given conditions (9), by the the property of similar triangles, ST×Ratio($\triangle$QBT,$\triangle$STE)=TQ (11);

Given conditions (3)(4)(10)(11), by the cyclic quadrilateral criterion (converse of the intersecting chords theorem), S, C, D, and Q are concyclic (12);

Given conditions (12), by the property of cyclic quadrilaterals (equal inscribed angles subtended by the same arc), we have $\angle QCD = \angle QSD$ (13);

Given conditions (9), by the property of similar triangles (equal angles), we have $\angle EST = \angle TQB$ (14);

By the triangle property (sum of interior angles equals 180°), we have $\angle PCQ = -\angle QPC - \angle TQB + 180$ (15);

By the theorem of adjacent supplementary angles, we have $\angle PCQ = -\angle QCD + 180$ (16);

By the common sense, $\angle EST = -\angle QSA + \angle QSD$ (17);

Given conditions ((13)(14)(15)(16)(17), by the cyclic quadrilateral criterion (based on angle relationships), P, R, Q, and S are concyclic (18);

Proof completed.

Figure 4: Human-like Solution (2022 IMO Problem 4).

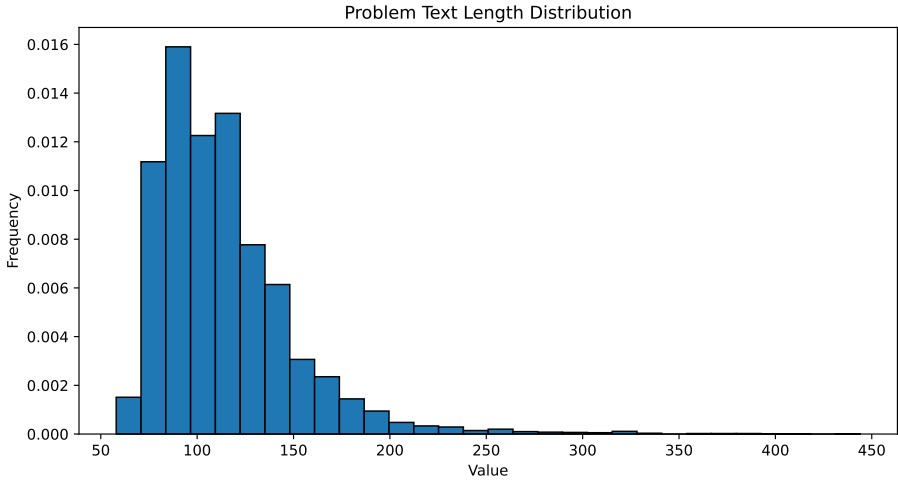

Figure 5: Problem Text Length Distribution.

## C   Statistics details

We collect information on the text length distribution, theorem length distribution, and predicate frequency in the formalgeo7k dataset.

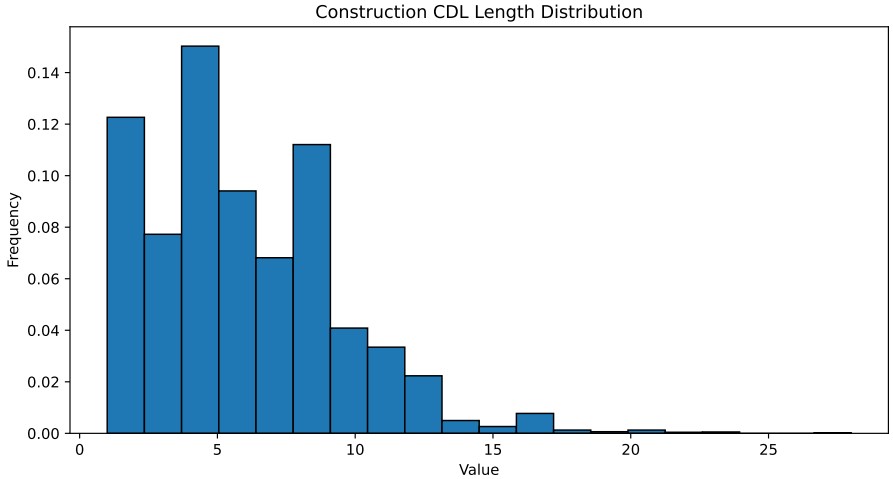

Figure 6: Construction CDL Length Distribution.

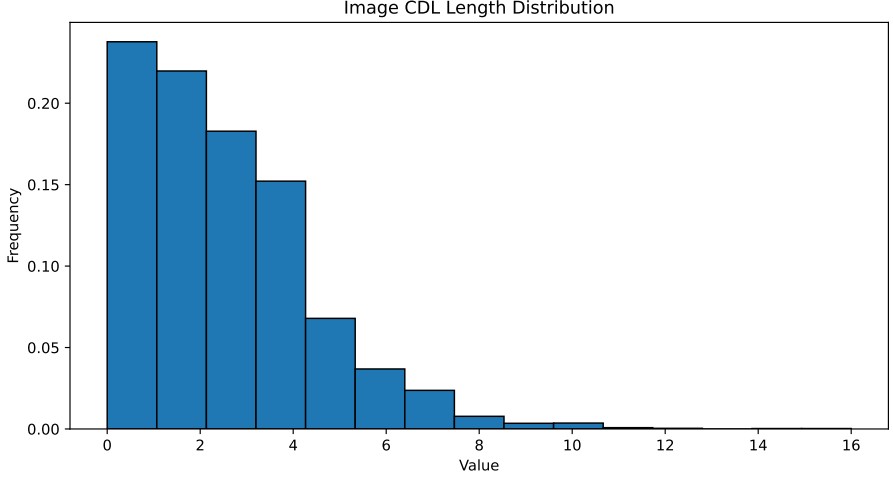

Figure 7: Image CDL Length Distribution.

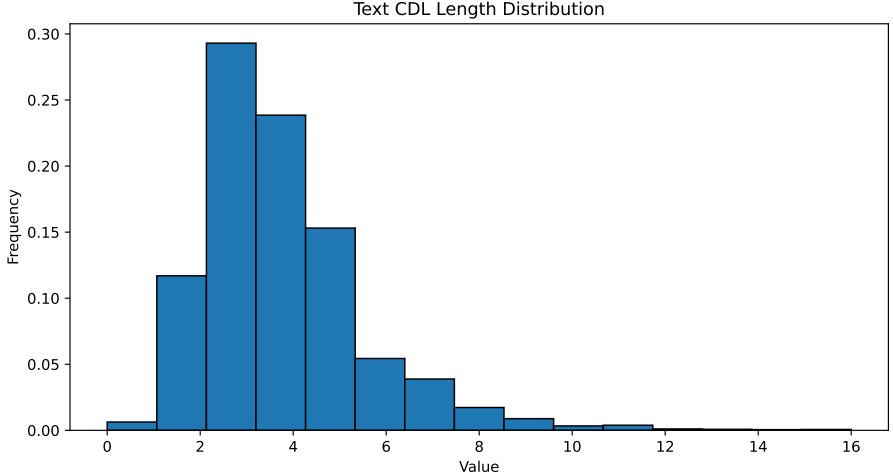

Figure 8: Text CDL Length Distribution.

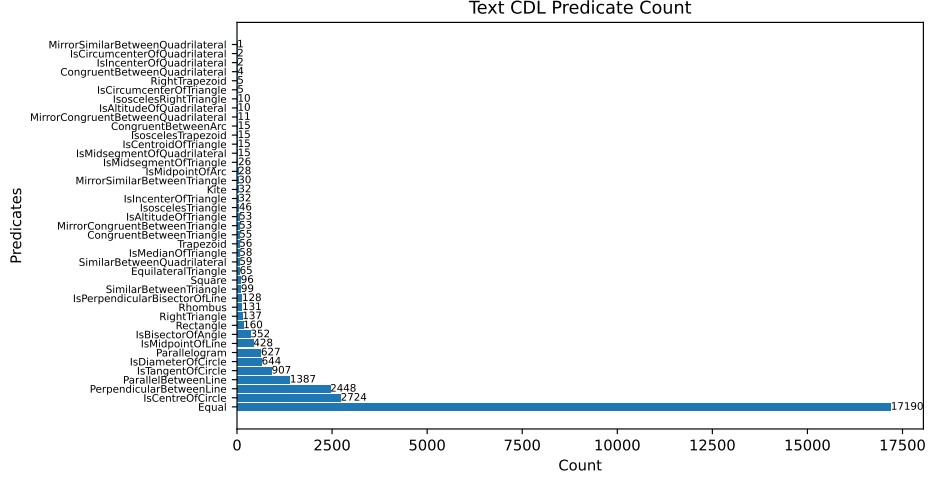

Figure 9: Text CDL Predicate Count.

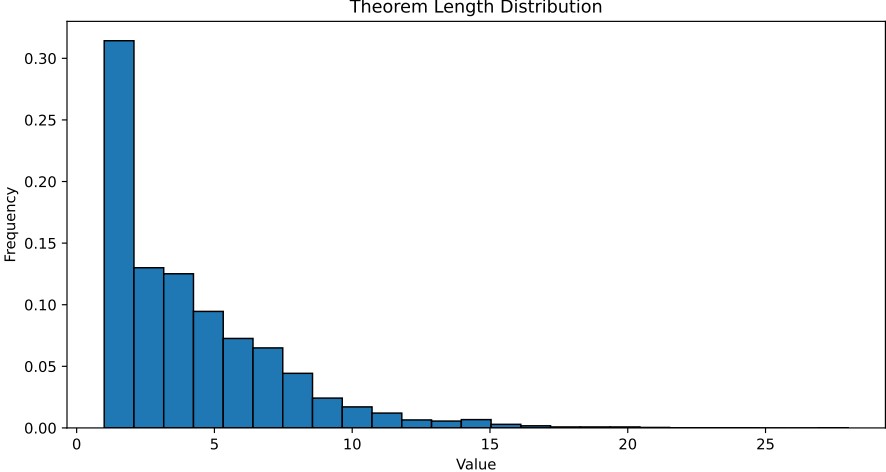

Figure 10: Theorem Length Distribution.

## D  Dataset download and Usage

You can use pip to download and use our formalgeo7k dataset.

```
$ conda create -n <your_env_name> python=3.10
$ conda activate <your_env_name>
$ pip install formalgeo

>>> from formalgeo.data import download_dataset, DatasetLoader
>>> from formalgeo.solver import Interactor
>>> from formalgeo.parse import parse_theorem_seqs

>>> download_dataset(dataset_name="formalgeo7k_v2",
                     datasets_path="your_datasets_path")
>>> dl = DatasetLoader(dataset_name="formalgeo7k_v2",
                       datasets_path="your_datasets_path")

>>> solver = Interactor(dl.predicate_GDL, dl.theorem_GDL)
>>> problem_CDL = dl.get_problem(pid=1)
>>> solver.load_problem(problem_CDL)
>>> for t_name, t_branch, t_para in parse_theorem_seqs(problem_CDL["theorem_seqs"]):
        solver.apply_theorem(t_name, t_branch, t_para)
>>> solver.problem.check_goal()

>>> from formalgeo.tools import show_solution
>>> show_solution(solver.problem)
```

