# OpenReview forum: "Formal Representation and Solution of Plane Geometric Problems"
_NeurIPS.cc/2024/Workshop/MATH-AI — MATH-AI 24_

### Official Review · Reviewer_etp8 · 2024-10-07
**Large dataset for geometric problem with a new representation method**

**Rating:** 6
**Confidence:** 3

**Review:**

The main contribution of this paper is the new geometric problem dataset, formalgeo7k. The dataset contains enough examples for LLM training. It also includes advanced geometry problems. Along with the new dataset, the authors also suggested a new representation method. I agree that the new representation method is more readable. The paper claimed that the new representation method creates consistent solver reasoning process. I can see how this might be the case however it would be better to have a comparison metrics on this. In the experiment section, I appreciate that the authors provided the success rate vs. theorem sequence length. It would be nice to see the performance sliced by other factors. For table 2, it would be better if the actual performance on each tasks for each dataset is provided instead of a check mark.

Overall I think the paper still contribute a lot to the field for the nicely annotated dataset. The experiment section could be more detailed and extended to better prove the benefits of the new representation method. However, it does provide enough info for potential users of the datasets.

---

### Official Review · Reviewer_qKdW · 2024-10-07
**This paper  introduces formalgeo7k, a dataset designed to address challenges in geometric problem-solving through rigorous formalization and a consistent geometry formal system. It serves as a benchmark for tasks like geometric diagram parsing and problem-solving.**

**Rating:** 6
**Confidence:** 4

**Review:**

Pros:
1. A large dataset of geometry problems is provided in this paper, which may benefit for this field.

2. It demonstrates detailed comparisons between several deductive methods on this dataset, which may offer valuable insights for other researchers.

Cons:
1. This dataset is only composed of several existing easy datasets, and more difficult and diverse geometry datasets are required in this field.

2. Novel proving methods should be presented not just evaluating existing deductive engines.

3. Limited technical contributions.

---

### Official Review · Reviewer_wnJS · 2024-10-07
**Review of formalgeo-7k**

**Rating:** 8
**Confidence:** 3

**Review:**

The paper introduces FormalGeo, a consistent geometry formal system grounded in rigorous geometry formalization theory. Building upon this framework, the authors present formalgeo7k, a dataset comprising 7,000 plane geometric problems annotated with problem texts, diagrams, formal descriptions, and solutions. The dataset aims to serve as a benchmark for tasks such as geometric diagram parsing and geometric problem solving (GPS).

It is an interesting work and formalgeo-7k can be a significant contribution to evaluating VLMs and other models in solving geometry problems as it has a decent amount of data points and provides human-like solutions.

My concerns about this work are the clarity of metrics used and the motivation of the work. For instance, While the paper mentions that annotations were reviewed for correctness, it does not specify the criteria or the inter-annotator agreement metrics. Information about the quality control measures would add credibility to the dataset. Furthermore, there is no mention of whether the dataset, the FormalGeo system, or the symbolic solver FGPS are publicly available. Accessibility is crucial for a benchmark dataset to facilitate widespread research and validation. Also, while the authors mentioned that this dataset can be used to evaluate LLMs, they have not tested these models. Finally, the authors can better motivate their work by providing useful insights driven by testing different models on their dataset.

While this paper is a good contribution to the workshop, in future venues and conferences, I would like to see how humans perform against formalgeo-7k, how comprehensible the solutions are for humans, how clear and accurate its annotations are, and how different metrics available in this dataset (as mentioned in Table 2.) can be used to gain detailed insights into the relative strength of used models.

---

### Official Review · Reviewer_gTmV · 2024-10-09
**The paper introduces a new unified dataset for geometric problems field**

**Rating:** 7
**Confidence:** 4

**Review:**

The paper presents a valuable contribution to the field of geometric problem-solving by providing a large, well-annotated dataset and demonstrating its potential as a challenging benchmark.

Pros:
+ The paper acknowledges and addresses the gap of missing symbolism in existing datasets and builds a unified benchmark.
+ The authors also use FGPS to verify the correctness of the syntax
+ The paper details how the dataset is constructed.

Cons:
- The paper doesn’t discuss the potential reason why existing works don’t perform well on the new dataset.
- It would be great if tasks such as GPS, GDP, and GRE can be evaluated individually against formalgeo7k to demonstrate the effectiveness of the new dataset.

---

### Decision · Program_Chairs · 2024-10-09

Accept